# A Workflow for Offline Model-Free Robotic Reinforcement Learning

**Aviral Kumar**[⋆,1]**, Anikait Singh**[⋆,1]**, Stephen Tian**[1]**, Chelsea Finn**[2]**, Sergey Levine**[1]

[1] UC Berkeley, [2] Stanford University     (* Equal Contribution)

`aviralk@berkeley.edu, asap7772@berkeley.edu`

**Abstract:** Offline reinforcement learning (RL) enables learning control policies by utilizing only prior experience, without any online interaction. This can allow robots to acquire generalizable skills from large and diverse datasets, without any costly or unsafe online data collection. Despite recent algorithmic advances in offline RL, applying these methods to real-world problems has proven challenging. Although offline RL methods can learn from prior data, there is no clear and well-understood process for making various design choices, from model architecture to algorithm hyperparameters, without actually evaluating the learned policies online. In this paper, our aim is to develop a practical workflow for using offline RL analogous to the relatively well-understood workflows for supervised learning problems. To this end, we devise a set of metrics and conditions that can be tracked over the course of offline training, and can inform the practitioner about how the algorithm and model architecture should be adjusted to improve final performance. Our workflow is derived from a conceptual understanding of the behavior of conservative offline RL algorithms and cross-validation in supervised learning. We demonstrate the efficacy of this workflow in producing effective policies without any online tuning, both in several simulated robotic learning scenarios and for three tasks on two distinct real robots, focusing on learning manipulation skills with raw image observations with sparse binary rewards. Explanatory video and additional content can be found at sites.google.com/view/offline-rl-workflow.

**Keywords:** workflow, offline RL, offline tuning

## 1 Introduction

Offline reinforcement learning (RL) can in principle make it possible to convert existing large datasets of robotic experience into effective policies, without the need for costly or dangerous online interaction for each training run. While offline RL algorithms have improved significantly [1, 2, 3, 4, 5], applying such methods to real-world robotic control problems presents a number of major challenges. In standard online RL, any intermediate policy found during training is executed in the environment to collect more experience, which naturally allows for an evaluation of the policy performance. This ability to evaluate intermediate policies lets practitioners use "brute-

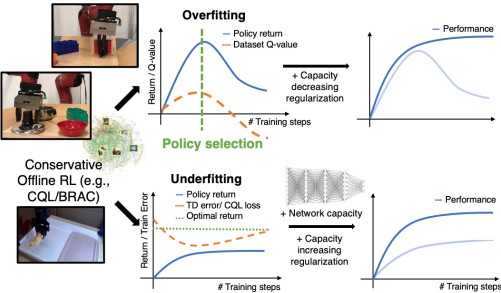

Figure 1: **Our proposed workflow** aims to detect overfitting and underfitting, and provides guidelines for addressing these issues via policy selection, regularization, and architecture design. We evaluate this workflow on **two** real-world robotic systems and simulation domains, and we find it to be effective.

force" to evaluate the effects of various design factors, such as model capacity and expressivity, the number of training steps, and so forth, and facilitates comparatively straightforward tuning. In contrast, offline RL methods do not have access to real-world on-policy rollouts for evaluating the learned policy. Thus, in order for these methods to be truly practical for real-world applications, we not only require effective algorithms, but also an effective *workflow*: a set of protocols and metrics that can be used to reliably and consistently adjust model capacity, regularization, etc in offline RL to obtain policies with good performance, without requiring real-world rollouts for tuning.

5th Conference on Robot Learning (CoRL 2021), London, UK.

A number of prior works have studied *model selection* in offline RL by utilizing off-policy evaluation (OPE) methods [6] to estimate policy performance. These methods can be based either on model or value learning [7, 8, 9, 10] or importance sampling [6, 11, 12, 13]. However, developing reliable OPE methods is itself an open problem, and modern OPE methods themselves suffer from hyperparameter selection challenges (see Fu et al. [14] for an empirical study). Moreover, accurate off-policy evaluation is likely not necessary to simply tune algorithms for best performance – we do not need a precise estimate of *how* good our policy is, but rather a workflow that enables us to best improve it by adjusting various algorithm hyperparameters.

In this paper, we devise a practical workflow for selecting regularizers, model architectures, and policy checkpoints for offline RL methods in robotic learning settings. We focus on a specific class of conservative offline RL algorithms [15, 2] that regularize the Q-function, but also show that our workflow can be effectively applied to policy constraint methods [16]. Our aim is not to focus on complete off-policy evaluation or to devise a new approach for off-policy evaluation, but rather to adopt a strategy similar to the one in supervised learning. Analogously to how supervised learning practitioners can detect overfitting and underfitting by tracking training and validation losses, and then adjust hyperparameters based on these metrics, our workflow (see Figure 1 for a schematic) first defines and characterizes overfitting and underfitting, proposes metrics and conditions that users can track to determine if an offline RL exhibits overfitting or underfitting, and then utilizes these metrics to inform design decisions pertaining to neural net architectures, regularization, and early stopping. This protocol is intended to act as a "user's manual" for a practitioner, with guidelines for how to modify algorithm parameters for best results without real-world evaluation rollouts.

The primary contribution of this paper is a simple yet effective workflow for robotic offline RL. We propose metrics and protocols to assist practitioners in selecting policy checkpoints, regularization parameters, and model architectures for conservative offline RL algorithms such as CQL [2] and BRAC [16]. We empirically verify the efficacy of our proposed workflow on simulated robotic manipulation problems as well as three real-world robotic manipulation problems on two different robots, with diverse objects, pixel observations, and sparse binary reward supervision. Experimentally, we evaluate our method on two real-world robots (the Sawyer and WidowX robots), and one realistic simulated tasks. Our approach is effective in all of these cases, and on two tasks with the Sawyer robot that initially fail completely, our workflow improves the success rate to **70%**.

## 2  Preliminaries, Background, and Definitions

The goal in RL is to optimize the infinite horizon discounted return $R = \sum_{t=0}^{\infty} \gamma^t r(\mathbf{s}_t, \mathbf{a}_t)$, where $r(s, a)$ represents the reward function evaluated at a state-action pair $(\mathbf{s}, \mathbf{a})$. We operate in the *offline* RL setting and are provided with a fixed dataset $\mathcal{D} = \{(\mathbf{s}, \mathbf{a}, r(\mathbf{s}, \mathbf{a}), \mathbf{s}')\}$, consisting of transition tuples obtained from rollouts under a behavior policy $\pi_\beta(\mathbf{a}|\mathbf{s})$. Our goal is to obtain the best possible policy by only training on this fixed offline dataset $\mathcal{D}$, with no access to online rollouts. We focus on *conservative* offline RL algorithms that modify the Q-function to penalize distributional shift, with most experiments on CQL [2], though we also adapt our workflow to BRAC [16] in Appendix F.1.

**Conservative Q-learning (CQL).** The actor-critic formulation of CQL trains a Q-function $Q_\theta(\mathbf{s}, \mathbf{a})$ with a separate policy $\pi_\phi(\mathbf{a}|\mathbf{s})$, which maximizes the expected Q-value $\mathbb{E}_{\mathbf{s}\sim\mathcal{D}, \mathbf{a}\sim\pi_\phi}[Q_\theta(\mathbf{s}, \mathbf{a})]$ like other standard actor-critic deep RL methods [17, 18, 19]. However, in addition to the standard TD error $\mathcal{L}_{\text{TD}}(\theta)$ (in blue below), CQL applies a regularizer $\mathcal{R}(\theta)$ (in red below) to prevent overestimation of Q-values for out-of-distribution (OOD) actions. This term minimizes the Q-values under a distribution $\mu(\mathbf{a}|\mathbf{s})$, which is automatically chosen to pick actions $\mathbf{a}$ with high Q-values $Q_\theta(\mathbf{s}, \mathbf{a})$, and counterbalances this term by maximizing the values of the actions in the dataset:

$$\min_\theta \ \alpha \left( \mathbb{E}_{\mathbf{s}\sim\mathcal{D}, \mathbf{a}\sim\mu(\cdot|\mathbf{s})}\left[Q_\theta(\mathbf{s}, \mathbf{a})\right] - \mathbb{E}_{\mathbf{s}, \mathbf{a}\sim\mathcal{D}}\left[Q_\theta(\mathbf{s}, \mathbf{a})\right]\right) + \frac{1}{2}\mathbb{E}_{\mathbf{s}, \mathbf{a}, \mathbf{s}'\sim\mathcal{D}}\left[\left(Q_\theta(\mathbf{s}, \mathbf{a}) - \mathcal{B}^\pi\bar{Q}(\mathbf{s}, \mathbf{a})\right)^2\right], \quad (1)$$

where $\mathcal{B}^\pi\bar{Q}(\mathbf{s}, \mathbf{a})$ is the Bellman backup operator with a delayed target Q-function, $\bar{Q}$: $\mathcal{B}^\pi\bar{Q}(\mathbf{s}, \mathbf{a}) := r(\mathbf{s}, \mathbf{a}) + \gamma\mathbb{E}_{\mathbf{a}'\sim\pi(\mathbf{a}'|\mathbf{s}')}[\bar{Q}(\mathbf{s}', \mathbf{a}')]$. In practice, CQL computes $\mu(\mathbf{a}|\mathbf{s})$ using actions sampled from the policy $\pi_\phi(\mathbf{a}|\mathbf{s})$. More discussion of CQL is in Appendix B. In this paper, we will utilize CQL as a base algorithm that our workflow intends to tune, but we also extend it to BRAC.

**Overfitting and underfitting in CQL.** Conservative offline RL algorithms [2, 20] like CQL can be sensitive to design choices, including number of gradient steps for training [21, 22] and network capacity. These challenges are also present in supervised learning, but supervised learning methods benefit from a simple and powerful workflow that involves using training error and validation error to characterize overfitting and underfitting. A practitioner can then make tuning choices based on

these characterizations. To derive an analogous workflow for offline RL, we first ask: **what do overfitting and underfitting actually mean for the case of conservative offline RL?**

To define overfitting and underfitting generically for any conservative offline RL method, we consider an abstract optimization formulation for such methods [2]:

$$\pi^* := \arg\max_{\pi} \; J_{\mathcal{D}}(\pi) - \alpha D(\pi, \pi_\beta) \qquad \text{(Conservative offline RL)}. \qquad (2)$$

$J_{\mathcal{D}}(\pi)$ denotes the average return of policy $\pi$ in the empirical MDP induced by the transitions in the offline dataset $\mathcal{D}$, and $D(\pi, \pi_\beta)$ denotes a closeness constraint to the behavior policy, effectively applied by the offline RL method. Our definition of conservative offline RL requires that this

| Quantity | Supervised Learning | Conservative Offline RL |
|---|---|---|
| Test error | Loss $\mathcal{L}$ evaluated on test data, $\mathcal{D}_{\text{test}}$ | Performance of policy, $J(\pi)$ |
| Train error | Loss $\mathcal{L}$ evaluated on train data, $\mathcal{D}_{\text{train}}$ | Objective in Equations 2, 1 |
| Overfitting | $\mathcal{L}(\mathcal{D}_{\text{train}})$ low, $\mathcal{L}(\mathcal{D}_{\text{val}})$ high, $\mathcal{D}_{\text{val}}$ is a validation set drawn i.i.d. as $\mathcal{D}_{\text{train}}$ | Training objective in Equation 1 is extremely low, low value of $J(\pi)$ |
| Underfitting | high value of train error $\mathcal{L}(\mathcal{D}_{\text{train}})$ | Training objective in Equation 1 is extremely high, low value of $J(\pi)$ |

Table 1: Summary of train error, test error and our definitions of overfitting and underfitting in supervised learning and conservative offline RL methods. We will propose metrics to measure these phenomena in a purely offline manner and recommend how to tune the underlying method accordingly.

divergence be computed in expectation over the state visitation distribution of the learned policy $\pi$ in the empirical MDP as discussed in Appendix F.1. For example, Equation 1 translates to utilizing $D_{\text{CQL}}(p, q) := \sum_{\mathbf{x}} p(\mathbf{x})(p(\mathbf{x})/q(\mathbf{x}) - 1)$ in Equation 2 (see Theorem 3.5 in Kumar et al. [2] for a proof). The training loss is discussed in Equations 1 and 2 and the test loss is equal to the negative of the actual return $J(\pi)$ of the learned policy. Analogously to supervised learning, we can use the notion of train and test error to define overfitting and underfitting in offline RL, as discussed in Table 1. However, note that the conditions summarized in Table 1 are not measurable completely offline. Precisely estimating if a run of an offline RL method overfits or underfits requires evaluating the learned policy via interaction with the real-world environment. In Section 3, our goal will be to devise offline metrics for characterizing overfitting that do not have this requirement. We will tailor our study specifically towards CQL, though we extend it to BRAC in Appendix F.1. A similar procedure could be devised for other offline RL methods, but we leave this for future work.

## 3 Detecting Overfitting and Underfitting in Conservative Offline RL

In standard supervised learning, we can determine if a method overfits or underfits by comparing the training loss to the same loss function evaluated on a held-out validation dataset, which serves as a "proxy" test dataset. In contrast, the return of the learned policy $J(\pi)$ in RL does not have a direct proxy that can be computed offline. Thus, our goal is to identify offline metrics and conditions that allow us to measure overfitting and underfitting in conservative offline RL, with a focus on CQL. We also adapt these conditions to BRAC [16], a policy-constraint method in Appendix F.2.

**Detecting overfitting in CQL.** Our definition of overfitting (Table 1) corresponds to a low value for the training loss (Equation 1), but poor actual policy performance $J(\pi)$. To detect this, we analyze the time series of the estimated Q-values averaged over the dataset *samples* $(\mathbf{s}, \mathbf{a}, r, \mathbf{s}') \in \mathcal{D}$ over the course of training with a large number of gradient steps. A run is labeled as overfitting if we see that the expected dataset Q-value exhibits a non-monotonic trend: if the average Q-values first increase and then decrease as shown in the figure on the right. Ad-

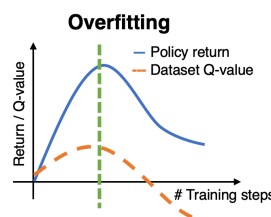

ditionally, we would see that training loss in Equation 1 eventually becomes very low. Why do we see such a trend in the average dataset Q-value? Since CQL selectively penalizes the average Q-value under the distribution $\mu(\mathbf{a}|\mathbf{s})$ supported on actions with large Q-values, we would expect the Q-values on states from the dataset $\mathbf{s} \sim \mathcal{D}$ and the learned $\mathbf{a} \sim \pi(\cdot|\mathbf{s})$ to be small since the policy is trained to maximize the Q-function as well. This in turn would lead to an eventual reduction in the average Q-value on dataset actions, $\mathbb{E}_{\mathbf{s}, \mathbf{a} \sim \mathcal{D}}[Q_\theta(\mathbf{s}, \mathbf{a})]$. This would be visible after sufficiently many steps of training, when values have propagated via Bellman backups in Equation 1 giving rise to the non-monotonic trend. If such a trend is observed, this raises two questions, as we discuss next.

***What does a low average Q-value*** $\mathbb{E}_{\mathbf{s}, \mathbf{a} \sim \mathcal{D}}[Q_\theta(\mathbf{s}, \mathbf{a})]$ ***imply about*** $J(\pi)$***?*** We show in Appendix A that, in principle, CQL training (Equation 1) should never learn Q-values smaller than the dataset Monte-Carlo return, and the Q-values should increase unless the learned policy $\pi$ is better than $\pi_\beta$. Intuitively, this is because the objective in Equation 1 aims to also *maximize* the average dataset

Q-value and thus the Q-values for the behavior policy are not underestimated in expectation. Now, if the policy optimizer finds a policy that attains a smaller learned Q-value than the dataset return, the policy can always be updated further towards the behavior policy so as to raise the Q-value. Therefore, Q-values can only decrease when the policy found by CQL is better than the behavior policy. We formalize this intuition in Appendix A in Theorem A.1. Thus, a low Q-value on $(\mathbf{s}, \mathbf{a}) \in \mathcal{D}$ indicates that the Q-function predicts extremely small Q-values on actions sampled from $\mu(\mathbf{a}|\mathbf{s})$. Typically, this would mean the highest Q-value actions $\mathbf{a}$ at a state $\mathbf{s} \in \mathcal{D}$ are those sampled from the offline dataset, drawn from the behavior policy. Thus, policy optimization, which aims to maximize the Q-value, would make $\pi(\mathbf{a}|\mathbf{s})$ closer to the behavior policy $\pi_\beta(\mathbf{a}|\mathbf{s})$ on $\mathbf{s} \in \mathcal{D}$.

***Which training checkpoint is likely to attain the best policy performance?*** Tracking overfitting in supervised learning is important for selecting the best-performing checkpoint, before overfitting becomes severe. Analogously, we can compare the average dataset Q-value across different checkpoints within the same run to pick the best policy. Since CQL aims to increase the average dataset Q-value (Equation 1), we would expect Q-values to initially increase, until learning starts to overfit and the average dataset Q-value starts decreasing. We should therefore select the latest checkpoint that corresponds to a peak in the estimated dataset Q-value. A visual illustration of this idea is shown in the figure on the previous page, where the checkpoint marked by the green line is recommended to be chosen. **In summary**, **(a)** to detect overfitting we can track:

> **Metric 3.1** (Overfitting). *A low average data Q-value $\mathbb{E}_{\mathbf{s},\mathbf{a}\sim\mathcal{D}}[Q_\theta(\mathbf{s},\mathbf{a})]$ that decreases with more gradient steps on Equation 1 indicates that the offline RL algorithm is overfitting.*

and **(b)** further, given a run that exhibits overfitting, our principle for policy selection is given by:

> **Guideline 3.1** (Policy selection). *If a run overfits (per Metric 3.1), select the checkpoint that attains the highest average dataset Q-value before overfitting for deployment.*

Finally, for actor-critic algorithms [18] that update the actor slower than the critic, the next policy checkpoint after the peak in the average dataset Q-value appears must be selected. In most of our experiments, we find that simply utilizing the policy checkpoint at the point of the peak in the Q-value also leads to good results making this a rare concern, but in some cases, utilizing the next checkpoint after the Q-value peak performs better empirically.

**Detecting underfitting in CQL.** Next, we turn to devising a procedure to detect underfitting. As summarized in Table 1, underfitting occurs when the RL algorithm is unable to minimize the training objective in Equation 1 effectively. Therefore, large values for the TD error, the CQL regularizer, or both imply underfitting. A large value for the CQL regularizer, $\mathcal{R}(\theta)$, indicates an overestimation of Q-values relative to their true value [2] and thus, unlike the overfitting regime, we would *not* expect the average learned Q-value to decrease with more training. Thus, one approach to predict underfitting is to track both the TD error, $\mathcal{L}_{\mathrm{TD}}(\theta)$, and the CQL regularizer, $\mathcal{R}(\theta)$, and check if the value of even one of these quantities is large. More discussion is provided in Appendix A.

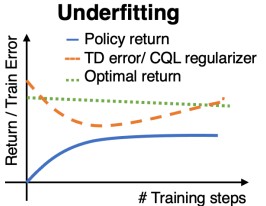

***How do we determine if the TD error and the CQL regularizer are "large"?*** In order to determine if the error of a particular run is large, we can rerun the base CQL algorithm but with models of higher capacity, which does not necessarily correspond to the function approximator size, as we will discuss in Section 4. For each model, we record the corresponding training errors and check if the training TD error and CQL regularizer value are reduced with capacity increase. If increasing capacity leads to a reduction in the loss without exhibiting the overfitting signs described previously, then we are in an underfitting regime. Another approach to answer the question is to utilize the value of the TD error ($\mathcal{L}_{\mathrm{TD}}(\theta)$) and the task horizon ($1/(1-\gamma)$) to estimate the overall error in the learned Q-values against the actual Q-value, which is equal to $\mathcal{L}_{\mathrm{TD}}(\theta)/(1-\gamma)$ [23] (see Appendix A). If this overall error spans the range of allowed Q-values on the task – which could be inferred based on the structure of the reward function in the task – then we can say that the algorithm is underfitting.

> **Metric 3.2** (Underfitting). *Compute the values of the training TD error, $\mathcal{L}_{\mathrm{TD}}(\theta)$ and CQL regularizer, $\mathcal{R}(\theta)$ for the current run and another identical run with increased model capacity. If the training errors reduce with increasing model capacity, the original run was underfitting.*

# 4 Addressing Overfitting and Underfitting in Conservative Offline RL

The typical workflow for supervised learning not only identifies overfitting and underfitting, but also guides the practitioner how to adjust their method so as to alleviate it (e.g., by modifying regularization or model capacity), thus improving performance. Can we devise similar guidelines to address overfitting and underfitting with conservative offline RL? Here, we discuss some ways to adjust regularization and model capacity to alleviate these phenomena.

**Capacity-decreasing regularization for overfitting.** As we observed in Section 3, the mechanism behind extremely low Q-values on the dataset is that CQL training minimizes Q-values on actions sampled from $\mu(\mathbf{a}|\mathbf{s})$. Two possible approaches to preventing over-minimization of these values are **(1)** applying regularization such as dropout [24] on Q-function layers, similar to supervised learning, and **(2)** enforcing that representations of the learned Q-function match a pre-specified target for all state-action tuples. For **(2)**, we can apply techniques such as a variational information bottleneck (VIB) [25, 26] regularizer on the learned representations, $\phi(\mathbf{s})$. Formally, let $(\mathbf{s}, \mathbf{a})$ denote a state-action pair. Instead of predicting a deterministic $\phi(\mathbf{s}) \in \mathbb{R}^d$ (Figure 10), we modify the Q-network to predict two distinct vectors, $\phi_m(\mathbf{s}) \in \mathbb{R}^d$ and $\phi_\Sigma(\mathbf{s}) \in \mathbb{R}^d$, and sample $\phi(\mathbf{s})$ randomly from a Gaussian centered at $\phi_m$ with covariance $\phi_\Sigma$, i.e., $\phi(\mathbf{s}) \sim \mathcal{N}(\phi_m(\mathbf{s}), \mathrm{diag}(\phi_\Sigma(\mathbf{s})))$. VIB then regularizes $\mathcal{N}(\phi_m(\mathbf{s}), \mathrm{diag}(\phi_\Sigma(\mathbf{s})))$ to be close to a prior distribution, $\mathcal{N}(0, \mathbb{I})$:

$$\min_\theta \ \mathcal{L}_{\mathrm{CQL}}(\theta) + \beta \mathbb{E}_{\mathbf{s} \sim \mathcal{D}} \left[ \mathrm{D_{KL}} \left( \mathcal{N}(\phi_m(\mathbf{s}), \mathrm{diag}(\phi_\Sigma(\mathbf{s}))) \, \middle|\middle| \, \mathcal{N}(0, \mathbb{I}) \right) \right] \quad \text{(VIB regularizer)}, \quad (3)$$

> **Guideline 4.1.** *To address overfitting, we recommend using some form of capacity-decreasing regularization on the Q-function, such as dropout or the VIB regularizer shown in Equation 3.*

**Capacity-increasing techniques for underfitting.** To address underfitting, we need to increase model capacity to improve optimization of the training objective. Analogous to supervised learning, model capacity can be increased by using more expressive neural nets (e.g., ResNets [27], transformers [28]) for representing the learned policy. We use ResNets in our experiments (Figure 10). However, the RL setting presents an additional challenge with capacity: while larger models *in principle* have more capacity, recent work [29, 21, 22] has shown that utilizing larger networks to represent Q-functions does not always improve its capacity in practice, because TD-based RL methods introduce an "implicit under-parameterization" effect that can result in aliased (i.e., similar) internal representations for different state-action inputs, even for very large neural networks that can express the true Q-function effectively. To address this issue, these works apply a "capacity-increasing" regularizer to Q-function training. For instance, we can use the DR3 regularizer [22], which penalizes the dot product of $\phi(\mathbf{s})$ and $\phi(\mathbf{s}')$ for a transition $(\mathbf{s}, \mathbf{a}, \mathbf{s}') \in \mathcal{D}$, and hence reduces aliasing. This objective is given by:

$$\min_\theta \ \mathcal{L}_{\mathrm{CQL}}(\theta) + \beta \mathbb{E}_{\mathbf{s}, \mathbf{a}, \mathbf{s}' \sim \mathcal{D}} \left[ \left| \phi(\mathbf{s})^\top \phi(\mathbf{s}') \right| \right] \quad \text{(DR3 regularizer [22])}, \quad (4)$$

> **Guideline 4.2.** *To address underfitting, we recommend using some capacity-increasing regularization on the Q-function and the policy either in conjunction or separately. Examples: **(1)** bigger policy networks (e.g., ResNets), **(2)** DR3 regularizer on the Q-network.*

# 5 Evaluation of Our Workflow Metrics and Protocols in Simulation

Next, we empirically validate the workflow proposed in Sections 3 and 4 on a suite of simulated robotic manipulation domains that mimic real-robot scenarios, from image observations with sparse binary rewards. We will examine how applying the workflow in Section 3 to detect overfitting or underfitting and then utilizing the strategies in Section 4 affects the performance of offline RL methods. An improved performance would indicate the efficacy of our workflow in making successful design decisions without any online tuning.

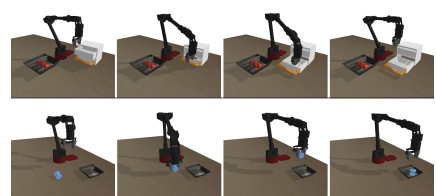

Figure 2: **Simulated domains [3] we use.**

**Experimental setup.** We use the environments from Singh et al. [3] to design offline RL tasks and datasets that we use for our empirical analysis. We consider two tasks: **(1)** a pick and place task and **(2)** a grasping object from a drawer task. Examples of trajectories in both of these simulated domains are shown in Figure 2 and are detailed in Appendix D. Briefly, the ***pick and place*** task consists of a 6-DoF WidowX robot in front of a tray with an object. The goal is to put the object inside the

tray. A non-zero reward of +1 is provided only when the object has been placed in the box. The offline dataset for this task consists of trajectories that grasp an object with a 35% success and other trajectories that place an object with a 40% success. Our second task is a ***grasping from drawer task*** where the WidowX robot is placed in front of a drawer and multiple objects. The robot can open or close the drawer, grasp objects from inside the drawer or on the table, and place them anywhere in the scene. The goal is to close the top drawer, then open the bottom drawer and take the object out. Only if the object has been taken out, a reward of +1 is obtained. The offline dataset consists of trajectories with a 30-40% success rate for opening and closing a drawer and other trajectories with only 40% placing success. We use $\alpha = 1.0$ for CQL training in all experiments, which is directly taken from prior work [3], without any tuning. However, too low or too high $\alpha$ values will inhibit the effectiveness of regular CQL and we first need to tune $\alpha$ as discussed in Appendix G. More details are provided in Appendix D.

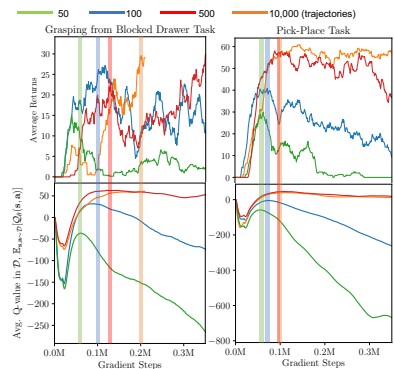

Figure 3: **Policy performance (Top) and average dataset Q-values of CQL (bottom) with varying number of trajectories.** Vertical bands indicate regions around the peak in average Q-value and observe that these regions correspond to policies with good actual performance.

**Scenario #1: Variable amount of training data.** Our first scenario consists of the simulated tasks discussed above with a variable number of trajectories in the training data (50, 100, 500, 10000). We run CQL and track metrics 3.1 and 3.2 in each case. Observe in Figure 3 (bottom) that with fewer trajectories, the average dataset Q-value $\mathbb{E}_{\mathbf{s},\mathbf{a}\sim\mathcal{D}}[Q_\theta(\mathbf{s},\mathbf{a})]$ first rises, and then drops. This matches the description of overfitting in Section 3. Observe in Figure 4 (left) that, at the same time, the value of the CQL regularizer is very low, which is not consistent with what we expect of underfitting. Thus, we can conclude that these conditions exhibit overfitting, especially with 50 and 100 trajectories. The vertical dashed lines indicate the checkpoints that would be selected for evaluation per Guideline 3.1. We further visualize the performance of the chosen checkpoints against the actual return of each intermediate policy in Figure 3 (top). Note that this value is obtained by rolling out the learned policy, and would not be available in a realistic offline RL setting, but is provided only for analysis. Selecting the checkpoint based on Guideline 3.1 leads us to select a model with close to the peak performance over the training process, validating the efficacy of Guideline 3.1.

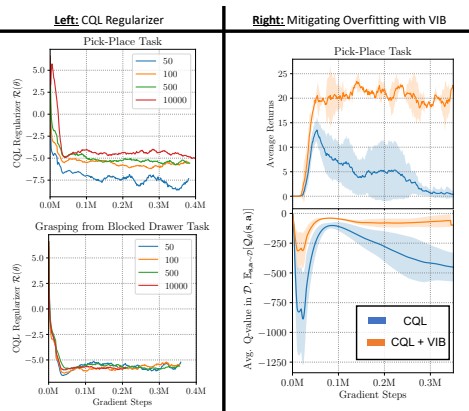

Figure 4: **Left:** CQL regularizer attains low values, especially with 50 and 100 trajectories in the pick and place task, **Right:** Using VIB mitigates overfitting, giving rise to a stable trend in Q-values and better performance which does not degrade with more training steps.

Since we detected overfitting by following our workflow, we now aim to address it by using the VIB regularizer in the setting with 100 trajectories. As shown in Figure 4 (right), applying this regularizer not only alleviates the drop in Q-values after many training steps, but allows us to pick later checkpoints in training which perform better than base CQL on both the tasks. This validates that overfitting, as detected via our workflow, can be effectively mitigated by decreasing capacity, in this case by using VIB. We evaluate dropout, $\ell_1$ and $\ell_2$ regularization schemes in Appendix J.

**Scenario #2: Multiple training objects.** Our second test scenario consists of the pick and place task, modified to include a variable number of object types (1, 5, 10, 20, 35). Handling more objects requires higher capacity, since each object has a different shape and appearance. In each case, CQL is provided with 5000 trajectories. Following our workflow from Section 3, we first compute the average dataset Q-value and the training TD error. We observe in Figure 5 that, unlike in Scenario #1, Q-values do not generally decrease when trained for many steps, suggesting that the Q-function is likely not overfitting. To check for underfitting, we visualize the training TD error and find that, with 10, 20 and 35 objects, TD error magnitudes are in the range of [1.0, 2.0], which suggests a

overall Q-value error of [30.0, 60.0] since the task horizon is 30. On an absolute scale, this error magnitude is large: since the rewards are 0/1, the range of difference between actual Q-values for any two policies is at most 30, which suggests that the error magnitude in the runs in Figure 5 are high. Hence, we conclude that this scenario generally exhibits underfitting with more objects. Indeed this trend is reflected in the policy performance that we plot for analysis in Figure 5: note that the policy return decreases with an increased number of objects, and the policy performance initially increases and saturates at a suboptimal value.

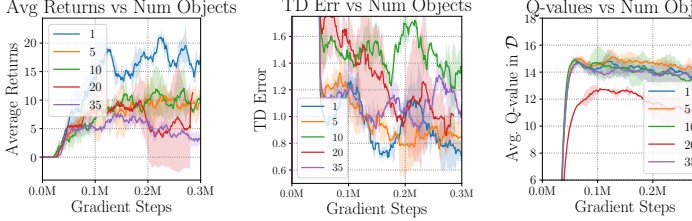 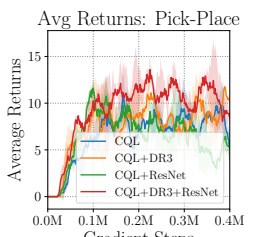

Figure 5: **Performance (left), TD error (middle) and average dataset Q-values (right) for the pick and place task with a variable number of objects.** Note that while the learned Q-values increase and stabilize, the TD error values in scenarios with more than 10 objects are large (1.0-2.0). Correspondingly, the performance generally decreases as the number of objects increases.

Figure 6: Correcting underfitting by applying our workflow for 35 objects.

To address underfitting in the multi-object case, we apply the proposed capacity-increasing measures to the 35-object task (results for 10 and 20 object settings are in Appendix I). We use a more expressive ResNet architecture for the policy and the DR3 regularizer for the Q-function together. Observe in the figure on the right that this combination (shown in red) improves policy performance in this setting (compared to green), which validates our workflow protocol for addressing underfitting.

## 6  Tuning CQL for Real-World Robotic Manipulation

Having evaluated the efficacy of our proposed workflow in simulation, we now utilize our workflow to tune CQL for real-world robotic manipulation. We test in two setups that require the robot to learn from sparse binary rewards and image observations. The settings differ in robot platform, task specification, and dataset size. Additional results and robot videos are at the following website: https://sites.google.com/view/offline-rl-workflow

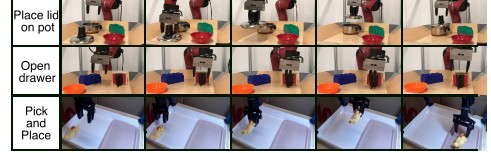

Figure 7: **Real-world tasks.** Successful rollouts of CQL tuned with our workflow from Sections 3 & 4. Top to bottom: Sawyer lid on pot, Sawyer drawer opening, WidowX pick-place task.

**Sawyer manipulation tasks [30].** First, we train a Sawyer robot in a tabletop setting to perform two tasks: **(1)** placing the lid onto a pot and **(2)** opening a drawer. The robot must perform these tasks in the presence of visual distractor objects, as shown in Figure 7. We directly use the dataset of 100 trajectories for each task collected by Khazatsky et al. [30] for our experiments so as to mimic the real-world use case of leveraging existing data with offline RL. We use four-dimensional actions with 3D end-effector velocity control in $xyz$-space and 1D gripper open/close action. More details regarding the setup are provided in Appendix D.

We run default CQL on these tasks and track the average Q-value, TD error, and CQL regularizer value. As shown in Figure 8, the average Q-value does not decrease over training, and the TD error (and CQL regularizer shown in Appendix E.2) is large. Per our discussion in Section 3, this indicates underfitting. Following our guidelines from Section 4, we utilize a more expressive ResNet policy (Figure 10), which increases the number of total convolutional layers from 3 to 9. We observe that this reduces the values of both the TD error Figure 8 and CQL regularizer (Appendix E.2) on both tasks. We

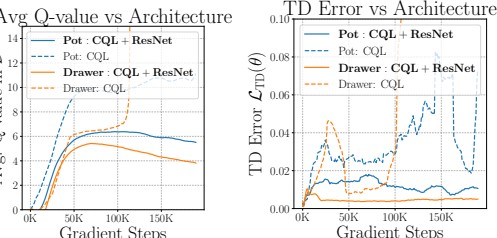

Figure 8: Average Q-value and TD error on Sawyer tasks as model capacity increases. Q-values increase over training with lower capacity ruling out overfitting and increasing model capacity leads to a reduction in TD error indicating the presence of underfitting.

then evaluate the learned policy over 12 trials conducted with different sets of distractor objects, including ones that are unseen during training. While the policy trained using base CQL is unable

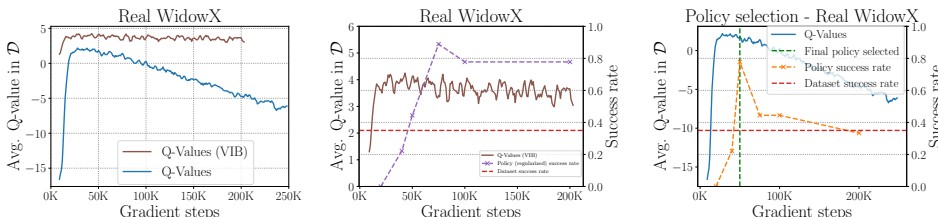

Figure 9: Q-values **(left)** and performance of CQL with **(middle)** and without **(right)** the variational information bottleneck correction for overfitting on the real-world widowX pick and place task. Since the Q-values start to decrease with more training, our workflow detects that CQL is overfitting. Using our policy selection guideline (Guideline 3.1) enables us to choose checkpoint 50 marked with the green vertical dashed line (right) which performs well. Further, addressing overfitting by applying the VIB regularizer stabilizes the Q-values (brown) which do not decrease unlike base CQL (blue) (left). Finally, applying the VIB regularizer improves performance and reduces sensitivity to policy selection (middle).

to successfully complete either task even once attaining a score of 0/12 on both tasks, the run that uses ResNet attains a significantly better success rate of **9/12** on the put lid on pot task and **8/12** on the drawer opening task, equal to **70.8%** success rate on average.

**WidowX pick and place task.** In our second setting, we tune CQL on a pick and place task with a WidowX 250 robotic arm, shown in Figure 7. The dataset consists of 200 trajectories collected by running a noisy scripted policy (Appendix D) with 35% success. We run CQL on this task and track the average Q-values, which we find initially increase and then decrease (Figure 9 (left; labeled as "Q-values")), indicating overfitting. We then evaluate our policy selection scheme, which in this case suggests deploying checkpoint 50, the immediate checkpoint after the peak in Q-values. To see if this checkpoint is effective, we evaluate the performance of a few other policy checkpoints (for analysis only) and plot this performance trend in Figure 9 (right) as a dashed line. Observe that indeed the checkpoint found by our workflow attains the highest success rate (**7/9**) compared to other checkpoints, which only succeed $\leq$ 4/9 times.

Since overfitting is detected, we now turn to addressing overfitting by adding the VIB regularizer (Equation 3) during training. As shown in Figure 9 (left), the Q-values obtained after the addition of this regularizer (shown in brown; labeled "Q-values (VIB)") are now stable and do not decrease over the course of training and so we can choose any policy for evaluation. We evaluate multiple policies, for visualization pur-

| Real-world WidowX pick and place | | | | | |
|---|---|---|---|---|---|
| **Method** | **Epoch** | 50 | 75 | 100 | 200 |
| CQL | | **7/9** | 4/9 | 4/9 | 2/9 |
| CQL + VIB | | 3/9 | **8/9** | 7/9 | 7/9 |

Table 2: Performance of various policy checkpoints of CQL and CQL + VIB on the real WidowX pick and place task (bold entry denotes the checkpoint selected by our workflow). Note that when overfitting is corrected via VIB, multiple checkpoints perform well.

poses only, in Figure 9 (middle), we find that all of them attain a $\geq$**7/9** success, comparable or better than the base CQL algorithm (Figure 9 (right)). This indicates that addressing overfitting not only leads to some gains in performance but also greatly simplifies policy selection as all checkpoints perform similarly and well. Table 2 summarizes these results below, where the bold entries denote the checkpoints found by our policy selection rule. These results indicate the effectiveness of our workflow in tuning CQL by addressing overfitting and underfitting on multiple real robot platforms.

## 7    Discussion

While offline RL algorithms have improved significantly, applying these methods to real-world robotic domains is still challenging due to little guidance on tuning them. In this paper, we devise a *workflow* for algorithms such as CQL and BRAC, which consists of a set of metrics and conditions that can be tracked by a practitioner over the course of offline training to detect overfitting and underfitting, and recommendations to addresses the observed challenges. Applying our workflow both in simulation and the real world shows strong performance benefits. While our proposed workflow is an initial step towards practical robotic offline RL and is based on our best conceptual understanding of certain offline RL algorithms, these guidelines are heuristic. To some extent this is unavoidable, since a workflow is a set of guidelines and recommendations, rather than a rigid algorithm. Regardless of how theoretically justified it is, in the end, its value is determined by its ability to produce good results. We believe the breadth of tasks considered, which consist of two different real robots and multiple simulated tasks, indicates its broad applicability. However, deriving theoretical guarantees regarding workflows of this type is an important direction for future research.

## Acknowledgements

We thank Ilya Kostrikov, Avi Singh, Ashvin Nair, Alexander Khazatsky, Albert Yu, Jedrzej Orbik, and Jonathan Yang for their help with setting up and debugging various aspects of the experimental setup as well as for providing us with offline datasets we could test our workflow on. We thank Dibya Ghosh, anonymous reviewers, and the area chair from CoRL for constructive feedback on an earlier version of this paper. AK thanks George Tucker and Rishabh Agarwal for valuable discussions. This research was funded by the DARPA Assued Autonomy Program and compute support from Google and Microsoft Azure.

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
