# OpenReview forum: "A Workflow for Offline Model-Free Robotic Reinforcement Learning"
_robot-learning.org/CoRL/2021/Conference — CoRL2021 Oral_

### Official Review · Reviewer_J1Lk · 2021-07-12

**Originality:** Good
**Technical Quality:** Good
**Clarity Of Presentation:** Very Good
**Impact:** 2

**Recommendation:**

Weak Accept: I recommend accepting the paper, but will not argue for my recommendation if the majority of other reviewers have a different opinion.

**Summary:**

Paper presents two metrics for detecting underfitting and overfitting in offline RL: 1) the magnitude of Q value 2) the TD-error vs. model size. The authors also provide suggestions to address these two issues in the light of the proposed metrics. Experiments on both simulated and real-world offline RL tasks demonstrate the effectiveness of the resulting model selection.

**Issues:**

1. More inspection on the decrease of Q value. Is overfitting the only possible factor that leads to this effect?

2. Is it possible to apply these two metrics to other offline RL algorithms?

**Reviewer Expertise:**

Good: General knowledge of the area

**Strengths And Weaknesses:**

[Strengths]

[+] The paper is overall clear and well-written. I like the illustrative writing style and I believe this paper is mostly readable to those who are not familiar with offline RL and related topics.

[+] The research problem seems to be incremental and it is indeed important -- model selection can be particularly difficult for offline RL as ideally no online eval should be allowed during training. The authors do bring intuitive and interesting perspectives to this problem.

[+] Although personally, I don't think this paper presents significantly novel ideas, the method here is well-motivated, reasonable, and technically sound.

[+] The empirical evaluations are impressive. I like how the authors extensively test their method on both simulated and real-world domains and attain consistent improvement over baselines.

[Weaknesses]

At this point, I cannot find any major issues with this paper. But I have to point out that the two metrics here are straightforward, especially for the underfitting one. The one on overfitting deserves more inspection as a decrease of Q value could also due to some other factors like weight decay.

As a side note, is it possible to extend these two metrics to other offline RL algorithms? I feel the first one highly relies on CQL.

**Summary Of Recommendation:**

Given the overall technical quality, I recommend an acceptance at this point. But I do hope the authors can address some of my concerns in the rebuttal and revise their paper accordingly.

---

> ### Author Response · Authors · 2021-08-29
> **Author Response**
>
> We thank the reviewer for the constructive feedback and for a  positive assessment of the paper.
>
> **Applicability to other offline RL algorithms:** We have now applied our overfitting workflow on a policy-constraint offline RL algorithm, BRAC (Wu et al. 2019), different from CQL. We provide a detailed response to this question in our [global comment to all the reviewers and the AC](https://openreview.net/forum?id=fy4ZBWxYbIo&noteId=7CIFzgxtrC) since this was a common question. We leave the details to the global comment and the section in **Appendices F.1 and F.2** and briefly, summarize our findings here. We find that in experiments with BRAC on the simulated drawer task with 100 and 200 trajectories from Scenario #1 in the paper, conservative Q-value estimates from BRAC can be used to detect overfitting and to perform policy selection, and dropout, a capacity-decreasing regularizer can be used for mitigating the overfitting phenomenon.
>
> We have also discussed in the revised paper (Appendix F.1) that our workflow is applicable to offline RL algorithms that can be expressed as optimizing the objective Equation 2, which we refer to as "conservative offline RL" algorithms.  Our workflow is not applicable to other offline RL algorithms such as TD3+BC, BEAR, AWR,  which do not fit the form of the conservative offline RL objective in Equation 2, which is required for our approach. Devising a workflow for these other algorithms is an interesting avenue for future work.
>
> ___
>
> **Regarding the decrease in Q-value,** we would note that in our experiments, we do not have any other active auxiliary loss such as weight decay during training. The Q-function is solely trained via the CQL training objective in Equation 1, and indeed the value of the training CQL regularizer is small in the simulated experiments in Scenario #1 where the algorithm overfits. Additionally, note that the decrease in Q-values is mitigated when a bigger dataset is used (see Figure 3, with 10000 trajectories), which is indicative of overfitting. Finally, we also performed an experiment applying $\ell_2$ regularization on the parameters of the Q-function in **Figure 20, Appendix I** and found that applying $l_2$ regularization to the parameters of the Q-function _mitigates_ the drop in Q-value and does not exacerbate it. If the reviewer has suggestions for any experiment, we are happy to include them in the final.

---

### Official Review · Reviewer_bymo · 2021-07-22

**Originality:** Good
**Technical Quality:** Good
**Clarity Of Presentation:** Very Good
**Impact:** 2

**Recommendation:**

Weak Accept: I recommend accepting the paper, but will not argue for my recommendation if the majority of other reviewers have a different opinion.

**Summary:**

Although current offline RL algorithms can learn from prior data without interacting with the environment, the learned policy has to be evaluated online. This paper proposes a workflow, including a set of protocols and metrics, to be used to adjust model capacity, regularization in offline model-free RL. The proposed workflow shows that it can successfully detect overfitting and underfitting in offline RL and thus can be used as guidelines to improve policy performance.


**Issues:**

The issues are listed above. Please check the "Strengths And Weaknesses".

**Reviewer Expertise:**

Good: General knowledge of the area

**Strengths And Weaknesses:**

Strengths:
1. The motivation is strong. Policy evaluation in offline RL is an important research topic. Although there are some limitations, the proposed workflow can be used to detect overfitting and underfitting in offline RL.
2. The paper offers well-designed experiments, both on simulators and real robots, to demonstrate the effectiveness of the workflow.
3. The paper is well-written and easy to follow.

Weaknesses:
1. My main concern is the effectiveness of the proposed metrics and the policy selection strategy.

    <a>  For metric 4.1 (overfitting), the decrease of the average data Q-value serves as the indicator of overfitting. But whether this metric is robust enough w.r.t other design/hyperparameter choices? For example, when choosing a larger \alpha, we may easily observe the decrease of Q-value. Also, when choosing an improper policy \mu(.|s), such as \mu(.|s)=behavior policy, Q-value may not decrease under overfitting.


    <b>  For guideline 4.1 (policy selection), the author suggests selecting the checkpoint that attains the highest average dataset Q-value before overfitting. But for example in Fig 9, the highest avg. Q-value is not corresponding to a good policy.

2. My second concern is whether the proposed workflow is limited to CQL (or Q-value regularized algorithms). In CQL, the Q function is trained with equation (1), but there are also many other offline RL methods, such as TD3_BC (https://arxiv.org/abs/2106.06860) etc., which add the constraint on policy while keeping the Q function untouched. In these algorithms, I’m afraid the proposed metrics are not suitable, e.g., even under overfitting, the Q function maybe won’t decrease like in CQL.

3. The proposed workflow, in my understanding, can only be used to detect overfitting and underfitting. But for the rest hyperparameters, for example, the \mu(.|s), \alpha in eq. (1), we still need to tune them via online evaluation.


**Summary Of Recommendation:**

The paper has several weaknesses as listed above, but considering the workflow can be successfully used to detect underfitting and overfitting as demonstrated in two different real-world robots and multiple simulated tasks, my recommendation is "Weak Accept".

---

> ### Author Response · Authors · 2021-08-29
> **Author Response**
>
> We thank the reviewer for their constructive feedback and for a positive assessment of the paper. We have updated the paper to address the main questions and detail our responses below in addition to the revision in the paper.
>
> ## Robustness of overfitting and policy selection strategies to $\alpha$ and $\mu(a|s)$ and tuning $\alpha$
>
> **Choice of $\alpha$:** We start by discussing the role of the CQL hyperparameter $\alpha$ in our workflow. In all our experiments, we utilized a single default value of $\alpha=1.0$ for the multiplier on the CQL regularizer drawn from prior work (Singh et al. 2020). Since this was a common question, we provide a detailed response in our [global comment to the Reviewers and the AC](https://openreview.net/forum?id=fy4ZBWxYbIo&noteId=dU6k2lKPEIF). To briefly summarize our findings, we found that our workflow is robust across several $\alpha$ values but fails to improve CQL when $\alpha$ is too large or too small. Note that CQL itself performs badly for these $\alpha$ values as an extremely small $\alpha$ fails to prevent catastrophic overestimation in Q-values and an extremely large $\alpha$ heavily regularizes the learned policy towards the behavior policy.  However, we have now updated the paper to propose two new guidelines **(Guidelines G.1 and G.2)** in Appendix G to handle these scenarios by first detecting if $\alpha$ is too small or too large and then modifying the value of $\alpha$ accordingly. After the value of $\alpha$ is modified, we can apply the workflow presented in the main paper to tune and improve the performance of CQL. Details including an empirical validation are in Appendix G, and our global comment is linked [here](https://openreview.net/forum?id=fy4ZBWxYbIo&noteId=dU6k2lKPEIF). These guidelines allow us to adjust the value of $\alpha$ fully offline.
>
> **Choice of $\mu(a|s)$:** With regards to the choice of $\mu(a|s)$, we would like to clarify that $\mu(a|s)$ is not a hyperparameter and we follow the exact same design choice of $\mu(a|s)$ as in CQL (Kumar et al. 2020) and COG (Singh et al. 2020) papers. Our aim is not to devise a new algorithm, but a workflow for adjusting regularization, capacity and early stopping for existing offline RL methods. To answer the reviewer’s question, in the setting when $\mu(a|s)$ is chosen to be the behavior policy, then the CQL regularizer would be $0$ in expectation and CQL would reduce to naive SAC. Such a SAC-style Q-function update no longer satisfies the functional form of conservative offline RL algorithms (Equation 2) where $\alpha > 0$, and hence is no longer covered by our workflow guidelines.
>
> Finally, we believe that the reason why the best performing policy in Figure 8 appears a little shifted from the peak in the Q-value is that the policy is updated slower than the Q-function in typical actor-critic algorithms and thus, the policy requires a few extra gradient steps to catch up to the Q-function. We will explicitly note this in the paper.
>
> ___
>
> ## Is the workflow limited to CQL or Q-function penalty methods?
>
> Our workflow is not limited to CQL or Q-function penalty methods but is applicable more generally to conservative offline RL methods that admit policy optimization objectives of the form of Equation 2 in the paper. To address this comment, we apply our workflow to BRAC (Wu et al. 2019), a policy constraint method that uses a policy optimization objective similar to the form of the objective in Equation 2. We provide a detailed response to this question in our [global comment to all the reviewers and the AC](https://openreview.net/forum?id=fy4ZBWxYbIo&noteId=7CIFzgxtrC) since this was a common concern. We leave the details to the global comment and the section in **Appendices F.1 and F.2** and briefly, summarize our findings here. We experiment with BRAC on the simulated drawer task with 100 and 200 trajectories and find that conservative Q-value estimates from BRAC can be used to detect overfitting and to perform policy selection. Then, dropout, a capacity-decreasing regularizer can be used for mitigating the observed overfitting phenomenon, which improves performance. This indicates that our workflow is not limited to CQL or Q-function penalty methods, and can be applied to other conservative offline RL methods such as BRAC as well.
>
> With regards to TD3+BC, we would like to first note that this algorithm was released on arXiv only 6 days before the CoRL deadline and hence it is concurrent to this paper. Second, we clarify that the set of conservative offline RL algorithms that fall in the scope of our workflow must be representable as optimization problems in Equation 2. TD3+BC can’t be represented in this formulation, and so our workflow is not directly applicable to this algorithm. We have added a discussion of this point in **Appendix F.1** and view devising workflow guidelines for TD3+BC as an interesting subject for future work. We also discuss this issue in detail in our global comment.

---

> > ### Comment · Reviewer_bymo · 2021-09-03
> > **Reply to authors**
> >
> > Thanks for the detailed answers and additional experiments. Also, sorry for using "hyperparameter" to describe the policy $\mu(a|s)$, it should be one design choice of CQL. After reading the author's response, I still have one question about the policy selection. In Fig 3, when training with 500 and 1000 trajectories, the curve Avg. Q is flat in both tasks while the Avg. returns vary a lot. This might be because of the environment, but I'm not sure whether a good policy can be properly selected by the proposed guideline. Also, guideline 4.1 says we should select the policy that corresponds to the highest average dataset Q-value, however, due to "the policy is updated slower than the Q-function in typical actor-critic algorithms", the best policy is not necessarily corresponding to the highest Q-value. Thus the effectiveness of the policy selection guideline is not very clear to me. Considering this, I will keep my recommendation as **weak accept**.

---

> > > ### Author Response · Authors · 2021-09-04
> > > **Author Response to Reviewer's Question**
> > >
> > > Dear Reviewer bymo,
> > >
> > > Thank you for the positive assessment of the work. We will make the paper more clear in regard to what to expect regarding the quality of the policies found via our guidelines. Our approach is definitely not guaranteed to find the best possible policy one could select if they had access to test return (which we don't claim to the best of our knowledge, and we will make this more explicit in the paper to avoid confusion), but rather serve as a guideline to obtain a good policy. This is similar to supervised learning where selecting the checkpoint based on accuracy on a held-out validation set may not select the checkpoint with the best test-set accuracy within the run. So you are broadly correct on both counts: in Fig 3 the very best policy is not selected (though a good policy is selected), and the lag between the critic and the actor can indeed lead to selecting a policy that is a bit worse than the best one. We will make sure to explicitly clarify this in the paper.
> > >
> > > That said, we would like to clarify the Fig 3 result -- we believe the reason why the policy selection guideline appears less effective in Figure 3 in the case of 500 trajectories, is because the plot is zoomed-out. Thus, to understand Figure 3 better, we provide a zoomed-in version of the plot for 500 trajectories and the corresponding policy performance in the figure at the [anonymous URL](https://imgur.com/a/D0WnFIL). Observe in this figure that our policy selection guideline does find a good policy, and the policy returns start to fluctuate after the peak in Q-values is obtained. Thus, our checkpoint selection rule may not find the _best_ policy checkpoint as evaluated under the test return, but our results show that it does find a good policy checkpoint based on test return. We will add this zoomed-in plot in the paper. Please let us know if this addresses your question.

---

> > > > ### Comment · Reviewer_bymo · 2021-09-04
> > > > **Reply to authors**
> > > >
> > > > Thanks so much for the explanation and the zoomed-in version of the plot for 500 trajectories in Fig. 3. I agree that the checkpoint selection rule may not find the best policy. However, in the zoomed-in plot, the checkpoint is selected at ~0.12M gradient steps, but at ~0.15M gradient steps, the avg.Q is also high (almost the same as the avg.Q at ~0.12M), and the policy drops obviously, which even can't be seen as a "good" policy. Why should we select the policy at ~0.12M rather than ~0.15M gradient steps, especially considering the lag between the critic and actor? This leads to my concern about the proposed guideline of policy selection: when the avg. returns are somehow noisy such as tasks in Fig 3 and Fig 9, if we don't access the test return, can we effectively use this guideline to select a "good" policy?

---

> > > > > ### Author Response · Authors · 2021-09-04
> > > > > **Response to Reviewer**
> > > > >
> > > > > The reason that the high peak at 0.15M is not selected is because our guideline is to pick the first peak before the average Q-values start to decrease, which happens at 0.12M. The blue band marked in the picture allows for the delay in the policy update beyond the first peak at 0.12M. But of course, you have a point that some variability/noise in learning could have resulted in the peak being later. As you also remarked, the reason this particular experiment is so difficult is that the true return varies with a very high frequency, making this a particularly difficult case for checkpoint selection. We will discuss this in the paper. In the other runs in Figure 3, we did not find the variability to be quite this high-frequency, and in that case, things are better behaved, but we acknowledge this is a limitation of this approach, and we will elaborate on it in the paper. There is certainly more work needed to develop more perfect workflows that are completely immune to this issue, and we hope that highlighting this difficult case will help other researchers address such limitations in future work.

---

### Official Review · Reviewer_D6Ls · 2021-07-22

**Originality:** Very Good
**Technical Quality:** Good
**Clarity Of Presentation:** Very Good
**Impact:** 3

**Recommendation:**

Strong Accept: I recommend accepting the paper and will argue for my recommendation even if other reviewers hold a different opinion.

**Summary:**

Offline reinforcement learning is very important to applications where data can be only acquired offline (e.g., medical application); or when one wants to avoid the application of untrained policy to real systems (e.g, robotic applications).

In supervised problems, there is a clear definition of "overfitting" and "underfitting".  More in detail, overfitting happens usually when the functional space of the approximator is large and there is a scarcity of samples. The result is that the model will "fit" the noise in the data. Such an issue can be mitigated by shrinking the functional space, introducing regularization, by the provision of more samples (when possible), or by early stopping. Underfitting happens usually when the functional space of the approximator is too small, or the regularization term is too strong, and the approximator will exhibit a high bias. Underfitting can be usually avoided by introducing a larger functional space or by weakening the effect of the regularization.

Eary stopping is a technique used along with gradient optimization techniques. The idea is that in the early stages of the optimization, the functional approximator is probably underfitting the function, while in the last stage of the learning procedure, the function could be fitting the noise. Therefore, if one could estimate when the overfitting starts, could stop the optimization at that point to avoid both the problems.
The detection of the underfitting or overfitting can be done by measuring the loss on the validation set. When the loss on the validation set starts to increase (and the loss of the training set keeps decreasing), we can detect the overfitting.

Most of the reinforcement learning literature focuses on online problems. Therefore, there is not a clear training set and validation set. Furthermore, the overfitting can be easily mitigated by the provision of new samples.

Contrarily, in offline reinforcement learning the training set is fixed.

This paper attempts to define overfitting and underfitting for offline reinforcement learning (focusing on conservative Q-Learning). Furthermore, since the most natural metrics would require interaction with the environment, the authors propose some heuristics to detect the underfitting or the overfitting without the interaction with the environment. Furthermore, they propose some techniques to overcome underfitting and overfitting.






**Issues:**

1) Metric 4.1.

In my opinion, the overfitting happens when the algorithm estimates a large return $\mathbb{E}_{\mu_0}[V_\theta(s)]$, while on the real environment the return starts actually to decrease.

I do understand that metric 4.1. detects sort-of when the maximization of the $Q$-function is already "over", and the regularization term starts to kick in. However, suppose that the network of the Q-function is _really_ small, and therefore overfitting cannot possibly happen.
Still, I can imagine that in that case, the metric 4.1. can still happen (when eq.1 reaches "convergence", and when the optimizer must exploit the regularization terms to increase the term in eq 2).

2) Metric 4.2.

Even though I generally accept the idea to train the dataset on two different functional approximators to detect the underfitting (if the one with higher capacity has a lower loss, then _probably_ the smaller one is underfitting); still I do not fully agree in the case of RL.
For example, can be that the smaller functional approximator is actually approximating the Q-function better, even though the empirical error is higher. In fact, it can happen that, if the Q-function has a higher magnitude, the magnitude of the error will also increase.

I know this effect, as I often noticed that at the beginning of the learning, the Bellman error tends to usually increase, as the magnitude of the Q-function also increases. That does not mean that the approximation is worst: actually, it is better, but the loss does not reflect it monotonically.

__side note__

I'd modify eq.2 with the following, for clarity,

$$
\pi^* := \mathrm{arg}\max_\pi J_D(\pi) - \alpha D(\pi, \pi_\beta)
$$

as $a$ and $s$ do not appear on the right-hand side of the equation.



**Reviewer Expertise:**

Good: General knowledge of the area

**Strengths And Weaknesses:**

__Strength__

The motivation of the paper is really strong. I believe that offline reinforcement learning is crucial in many applications. Furthermore, most of the literature on reinforcement learning does not mention the problem of overfitting or underfitting. The problem is even more important in offline reinforcement learning, as there is no possibility of providing more samples.
Measuring these quantities without the possibility of interacting with the environment is also very problematic.

For this reason, I believe that the attempt of the authors of providing a solution to this problem is very important for the community.

Even though, as I will write later, I am not fully convinced by the heuristics proposed in the paper, the experimental section is in my opinion strong enough to support the statements made.


__Weakness__


A first weakness of the paper is to consider the specific case of CQL. I do believe that overfitting or underfitting is a much larger and more general problem than a specific algorithmic solution such as CQL. I regard the regularization term in CQL as an attempt to avoid overfitting. The "surrogate" objective introduced in CQL, is a modification of the real objective that includes a regularization term. Therefore one should still focus on the real objective.

Another weakness of the paper is to look at the average $Q_\theta(s, a)$. I believe that a much better metric should be $J(\theta)$, which in terms of the algorithm can be estimated with $\mathbb{E}_{\mu_0}[V_\theta(s)]$. This is the metric that we want to optimize. Maximizing the average Q-function tells only how we are maximizing the surrogate objective. Refer to [1, 2, 3] to see why maximizing this objective is "wrong".

There are many forms of regularization. As said, I already believe that the regularization term in CQL is avoiding overfitting. Although I believe that VIB can be a good way to regularize the Q-function, I don't see why other simpler techniques have not been tested (L2, L1, ...).



[1] Section 5.1 of https://arxiv.org/pdf/1811.09013.pdf
[2] Section 3 paragraph "Training Mismatch" of http://proceedings.mlr.press/v97/fujimoto19a/fujimoto19a.pdf
[3] Section 4.1.2 of https://arxiv.org/pdf/2010.14771.pdf





**Summary Of Recommendation:**

I do recommend the acceptance of the paper, mostly because I think it is an important attempt to a problem that is really interesting for the community.
I am convinced that the paper is good, even though the proposed heuristics are relying on intuition, and I do not entirely believe that they are correct.

Update
==

I am satisfied with the author's answer and I am keeping my score unchanged.

---

> ### Author Response · Authors · 2021-08-29
> **Author Response: Part 1/2**
>
> We thank the reviewer for their constructive feedback and for a positive assessment of our paper. Below we address the weaknesses and questions raised by the reviewer:
>
> > **A first weakness of the paper is to consider the specific case of CQL**
>
> We provide a detailed response to this question in our [global comment to all the reviewers and the AC](https://openreview.net/forum?id=fy4ZBWxYbIo&noteId=7CIFzgxtrC) since this was a common concern. We leave the details to the global comment and the newly-added section in **Appendices F.1 and F.2** and briefly, summarize our findings here. Our workflow is applicable to offline RL algorithms that can be expressed as optimizing Equation 2, which we refer to as "conservative offline RL" algorithms. Policy constraint algorithms such as BRAC (Wu et al. 2019), also admit a similar optimization objective, and thus, **we applied our workflow to BRAC (Wu et al. 2019)**. We find that in experiments on the simulated pick and place tasks with 50 and 100 trajectories from Scenario #1, conservative Q-value estimates from BRAC can be used to detect overfitting and to perform policy selection. Dropout, a capacity-decreasing regularizer can be used for correcting this overfitting phenomenon and this leads to improved performance. We also clarify in the updated paper in Appendix F.1 that our workflow is not applicable for other offline RL algorithms such as TD3+BC, BEAR, AWR, which do not fit the form of the conservative objective in Equation 2, which is required for our approach. We have added this as an avenue for future work in the paper.
>
> ____
>
> > **Other schemes for correcting overfitting (L1, L2)**
>
> We have now added experiments in **Appendix I** that use other forms of capacity-decreasing regularization schemes for correcting overfitting in CQL. We experimented with three schemes: $\ell_1$, $\ell_2$ regularization on the parameters $\theta$ of the Q-function, and dropout on intermediate layers of the Q-function and found that $\ell_2$ regularization and dropout effectively mitigate overfitting in CQL on the simulated pick and place task from Scenario #1. As discussed in Appendix I, the hyperparameters of the regularization schemes (multiplier for $\ell_1$ and $\ell_2$ regularization and probability for dropout) were selected out of a candidate set of values in an offline fashion by picking the hyperparameter that mitigates the drop in Q-values, while maintaining a small value of the CQL regularizer.
>
> As shown in Figure 20 (left column) in Appendix I, adding dropout to the intermediate layers of the Q-function has a similar effect as adding the VIB regularizer: it mitigates a drop in the average dataset Q-value and improves the policy return. $\ell_2$ regularization also mitigates the drop in Q-values and improves performance (Figure 20 (middle column)). Our experiments applying $\ell_1$ regularization on the parameters of the Q-function either failed to mitigate the drop in average Q-value with more training steps or resulted in an inability to minimize the CQL regularizer. Erroneous overestimation as a result of imperfect minimization of the CQL regularizer likely caused the policy to choose out-of-distribution actions giving rise to poor performance.
>
> ___
>
> > **Average Q-value in the dataset $E\_{s, a \sim \mathcal{D}}[Q_\theta(s, a)]$ vs Q-value under the initial state distribution $E\_{s, a \sim \mu_0, \pi}[Q_\theta(s, a)]$**
>
> We implemented the metric suggested by the reviewer (average policy value under the initial state-distribution) and we found that this metric identical policy selection results as Metric 4.1 on the two simulated domains from Scenario #1. In experiments presented in **Appendix J (Figure 21)**, we find that both the metric proposed by the reviewer and Metric 4.1 evolve in an overlapping manner for the most part of training, and hence, gives rise to identical policy selection results. While we agree that the metric we use (expected dataset Q-value) is somewhat counterintuitive, in Appendix A.1 (Theorem A.1 & Lines 656-673) we present an argument based on a theoretical result in tabular settings that shows that quantity in Metric 4.1 (i.e. the average of the CQL-learned Q-values over the dataset, $E\_{s, a \sim \mathcal{D}}[Q_\theta(s, a)]$) should only decrease once the CQL algorithm has found a good policy that is better than the behavior policy. The intuition is the following: since CQL (Equation 1) directly maximizes the Q-values of dataset $(s, a)$ tuples (while minimizing OOD action values), if the Q-value at dataset actions is nonetheless decreasing, it means that the target values used to update the Q-values at dataset actions are very small, which should only happen when CQL is overfitting. We believe that the combination of the theoretical result and the good empirical performance of this criterion justifies its use in our work, though we will note that it is somewhat counterintuitive.

---

> > ### Author Response · Authors · 2021-08-29
> > **Author Response: Part 2/2**
> >
> > > **Issues: Metric 4.1**
> >
> > We would like to clarify the meaning of the term "overfitting" in this scenario, which we believe should resolve this question. Under our definition, overfitting is characterized by the scenario where the average Q-values on the dataset state-action tuples start to decrease with additional gradient steps and are generally low. This is the opposite of the reviewer’s interpretation of overfitting (“estimated return is larger than actual policy performance”) which corresponds to overestimation. We agree that, in principle, one could have defined overfitting as overestimation, but our nomenclature of overfitting does not refer to overestimation. We will include this clarification in the paper.
> >
> > To verify what happens when a really small Q-network is used, we ran an experiment where we utilize a very small Q-network (1 convolutional layer + 1 feed-forward layer) for CQL. In this case, we found that the average Q-value never decreased with more gradient steps, indicating the absence of overfitting per our definition in Metric 4.1. Instead, we observed that the CQL regularizer also attained positive values in this experiment, which suggests an imperfect minimization of out-of-distribution Q-values and thus an underfitting phenomenon (per Metric 4.2). Please let us know if this resolves the concern.
> >
> > ___
> >
> > > **Issues: Metric 4.2**
> >
> > We have updated the paper to clarify that the "capacity" of a model for the case of TD learning does not necessarily correspond to function approximator size. We agree with the reviewer that simply increasing the size/expressivity of the Q-function approximator may not lead to an improved fitting of the TD error and we discuss this issue in Section 5 (Lines 235-248). However, higher capacity can be attained by adding capacity-increasing regularization schemes (discussed in Lines 233-245 in the paper), which include DR3 (Anonymous et al. 2021), and we will note that increased model capacity in Metric 4.2 can be attained by adding these capacity-increasing regularizers onto the Q-function.

---

### Official Review · Reviewer_rBLy · 2021-07-24

**Originality:** Very Good
**Technical Quality:** Very Good
**Clarity Of Presentation:** Excellent
**Impact:** 4

**Recommendation:**

Strong Accept: I recommend accepting the paper and will argue for my recommendation even if other reviewers hold a different opinion.

**Summary:**

The paper discusses several practical considerations for conservative offline RL methods (CQL specifically). It provides metrics for practitioners to identify overfitting and underfitting of the algorithm without evaluating the policy with any online data. It also provides corresponding guidelines when overfitting or underfitting happens. The claims are supported by thorough experimental analysis on both simulated and real robots.


**Issues:**

Address the weaknesses mentioned above and provide more discussion on the extended questions.

**Reviewer Expertise:**

Very good: Comprehensive knowledge of the area

**Strengths And Weaknesses:**

Strengths:
- The issue discussed in the paper is important and also very common in offline RL when applied to robotics. The paper provides a thorough discussion on it and thus provides a strong contribution to the field.
- The claims are well supported by the analysis in the experiment section.
- The paper is well organized and easy to understand.

Weaknesses:
- Scenario #2: The capacity-increasing measures are evaluated using the 35-object task. From the policy performance plot and the TD-error plot on the website, the improvement of performance is not clear to me. A more extensive study is needed to verify the hypothesis of underfitting. Maybe it would be interesting to see the same experiments for the 10-object and 20-object tasks. Since they actually have larger TD errors than 35-object from Figure 5, we might see a clearer improvement.
- Line 172-175: “Thus, a low Q-value … on states in the training dataset.” Can you clarify this further?
- Figure on the right of Line 340-346: Missing figure number and description. The legend covers a part of the curves and should be adjusted.

Extended questions:
- How much does the CQL hyperparameter alpha affect the conclusions? For the overfitting case, would alpha affect the drop of the Q-value?
- For the robot experiment, how are the instability and underfitting related to overestimation?
- Line 415: “Our workflow is also specific to conservative offline RL methods, and particularly CQL.” It would be great if the authors can discuss this a bit more. To what extent do the conclusions hold for general offline RL or online RL methods?

**Summary Of Recommendation:**

The paper proposes metrics and guidelines for overfitting and underfitting issues of CQL. It provides a thorough analysis of the issues and solutions with experiments in simulation and in the real world with considerably large-scale and challenging manipulation tasks. The paper provides important insights for offline RL. It is both technically informative and enjoyable to read.

---

> ### Author Response · Authors · 2021-08-29
> **Author Response: Part 1/2**
>
> We thank the reviewer for their constructive feedback and for a positive assessment of our paper. We have added several experiments in the updated paper to address the comments and to answer the extended questions that we detail below.
>
> ### Questions
>
> 1. **Scenario #2:** To address the comment about capacity-increasing measures in Scenario #2, we have now updated the paper to include preliminary experiments with 10 and 20 objects in **Appendix H, Figure 19**. We observe that utilizing a larger ResNet policy + DR3 capacity-increasing regularizer to address underfitting leads to an improvement over naive CQL for both 10 and 20 objects, and gives rise to a smaller value of TD error. Notably, we find that the return increases from about 7.0 to 14.0 for the run with 10 objects. We will perform a more complete study for the final version of the paper.
>
> 2. **Lines 172-175:** We have edited the text to modify Lines 172-175 to make them more clear. These lines were used to indicate that typically when a run of CQL overfits, the Q-values at state-action tuples in the dataset $(s, a) \in \mathcal{D}$ are likely to be higher than the Q-values at $s \in \mathcal{D}$ but for other unseen actions sampled from $\mu(a|s)$. Since the learned policy $\pi$ aims to maximize the Q-function, it would then select the actions from the dataset (i.e., the behavior policy) since these actions have the highest Q-value at a given state. Hence, this would update the learned policy $\pi(a|s)$ towards the behavior policy $\pi_\beta(a|s)$.
>
> 3. We have updated the figure in Lines 340-346 with a caption and figure number (Figure 6 in the updated paper). We will incorporate a bigger version of the plot in the final version of the paper.
>
> ____
>
> ### Extended Questions
>
> 1. **How are instability and underfitting related to overestimation?:** This is a great question! In the real robot experiments, the inability to fit the training objective using naive CQL does lead to overestimated Q-values (seen in Figure 7, left), which is perhaps unsurprising since policy optimization would prefer to choose out-of-distribution actions with overestimated Q-values when the training CQL regularizer is not small enough. Thus, for the case of underfitting (i.e., a large value of training TD error and CQL regularizer), the policy optimization procedure is likely to find actions with erroneously high Q-values, and thus the value of the policy (discounted policy return) is likely going to be smaller than the learned Q-value, indicating the presence of overestimation. We empirically verify that overestimation happens in the real robot experiments (Figure 8, left indicates this since the policy return for naive CQL is essentially 0 as it fails to solve the task, but the Q-values for the run are still positive). Of course, note that overestimation cannot be utilized by itself within a workflow, since overestimation cannot be quantified without access to the ground truth return of the policy being learned.
>
> 2. **Applying our workflow to tune other offline RL algorithms beyond CQL:** We provide a detailed response to this question in our [global comment to all the reviewers and the AC](https://openreview.net/forum?id=fy4ZBWxYbIo&noteId=7CIFzgxtrC) since this was a common concern. We leave the details to the global comment and the section in **Appendices F.1 and F.2** and briefly, summarize our findings here. Our workflow is applicable to offline RL algorithms that can be expressed as optimizing Equation 2, which we refer to as "conservative offline RL" algorithms. Policy constraint algorithms such as BRAC (Wu et al. 2019), also admit a similar optimization objective, and thus, **we apply our workflow to BRAC (Wu et al. 2019) in Appendix F.2**. We find that in experiments on the simulated drawer task with 100 and 200 trajectories from Scenario #1 in the paper, conservative Q-value estimates from BRAC can be used to detect overfitting and to perform policy selection. Dropout, a capacity-decreasing regularizer can be used for mitigating the overfitting phenomenon and this improves the policy performance. We also now discuss in **Appendix F.1** in the updated paper, that our workflow is not applicable for other offline RL algorithms such as TD3+BC, BEAR, AWR, which do not fit the form of the conservative objective in Equation 2, which is required for our approach. Devising a workflow for these algorithms is an interesting avenue for future work.

---

> > ### Author Response · Authors · 2021-08-29
> > **Author Response: Part 2/2**
> >
> > ### Extended Questions (Continued)
> >
> > 3. **Effect of CQL hyperparameter $\alpha$ on conclusions:** In all our experiments, we utilized a single default value of $\alpha=1.0$ for the multiplier on the CQL regularizer drawn from prior work (Singh et al. 2020). Since this was a common question, we provide a detailed response in our [global comment to the Reviewers and the AC](https://openreview.net/forum?id=fy4ZBWxYbIo&noteId=dU6k2lKPEIF). To briefly summarize our findings, we found that our workflow is robust across several $\alpha$ values, but fails to improve CQL when $\alpha$ is too large or too small. Note that CQL itself performs badly for these $\alpha$ values as an extremely small $\alpha$ fails to prevent catastrophic overestimation in Q-values and an extremely large $\alpha$ heavily regularizes the learned policy towards the behavior policy.  However, we have now updated the paper with two new guidelines **(Guidelines G.1 and G.2)** in **Appendix G** to handle these scenarios by first detecting if $\alpha$ is too small or too large and then modifying the value of $\alpha$ accordingly. After the value of $\alpha$ is modified, we are able to apply the workflow presented in the main paper to tune and improve the performance of CQL. Details including an empirical validation are provided in Appendix G, and our global comment is linked [here](https://openreview.net/forum?id=fy4ZBWxYbIo&noteId=dU6k2lKPEIF).

---

> > > ### Comment · Reviewer_rBLy · 2021-09-03
> > > **Reply to authors**
> > >
> > > Thank the authors for the additional discussion and experimental results. I believe the contribution of the paper is solid and I found it informative to read. I will keep my ratings.

---

### Author Response · Authors · 2021-08-29
**Answers to Common Questions of the Reviewers and the AC (Part 1/2)**

In this global comment, we address some questions asked by multiple reviewers pertaining to: **(1)** applying our workflow to other offline RL algorithms and **(2)** the effect of the CQL hyperparameter $\alpha$ on our workflow. We have added sections in the Appendix **(Appendices F and G)** which can be accessed at the [anonymous URL here](https://drive.google.com/file/d/1OeHoJmRxZqMKePEOjQUSIlhlvPrQWPhZ/view?usp=sharing) or can be found in the supplementary material.

## 1. Applying Our Workflow to BRAC, a policy-constraint offline RL algorithm

To assess the applicability of our workflow to other offline RL algorithms, we use our workflow to tune BRAC (Wu et al. 2019), a policy constraint offline RL method that satisfies our definition of conservative offline RL algorithms in Equation 2. We ran experiments on the simulated grasping from drawer task in Scenario #1 (Figure 2) with 100 and 200 training trajectories to assess if our workflow for detecting overfitting, policy selection, and overfitting correction are applicable to BRAC. We present a brief summary of our findings below and we have also added the discussion to **Appendix F.2** in blue, along with empirical results. A summary of how our workflow for CQL can be adapted to BRAC is shown in **Table 2 in Appendix F**.

**Overfitting detection and policy selection guidelines for BRAC (Table 2):** Unlike CQL where the learned Q-network directly predicts a conservative Q-estimate, which accounts for both the reward and the divergence from the behavior policy, BRAC estimates these quantities separately. Specifically, the Q-values learned by BRAC do not account for the divergence against the behavior policy at the current state. Therefore, to be able to apply our workflow guidelines (Metric 4.1, Guideline 4.1) to BRAC, we need to track a "conservative Q-value estimate" which is used by BRAC for policy optimization and is also equal to the target value used in the BRAC Bellman update. This conservative estimate is formally given by:  $Q_c(s, a) := Q(s, a) + \beta \log \hat{\pi}_\beta(a|s)$, where $\hat{\pi}_\beta$ denotes a learned model of the behavior policy and $\beta$ denotes the coefficient of behavior regularization. A complete discussion of the BRAC training objectives is provided in Equation 9 in Appendix F.1 of the revised paper. As shown in the plot in **Figure 14 (left)** in Appendix F.2, $\mathbb{E}\_{s, a \sim \mathcal{D}}[Q_c(s, a)]$ first increases and then decreases with 100 trajectories, whereas it is relatively stable for 200 trajectories. This indicates the presence of overfitting in the run with 100 trajectories. As shown via vertical bars in Figure 14 (left), the policy checkpoints near the peak in conservative Q-values correspond to the high-performing policy checkpoints within the run. This indicates that our workflow for policy selection (Guideline 4.1) is effective for BRAC as well.

___

**Addressing overfitting for BRAC:** Next, we test the efficacy of Guideline 5.1 by correcting overfitting for the BRAC run with 100 trajectories. We need to apply a capacity-decreasing regularizer to mitigate overfitting. This regularizer must be applied on the conservative Q-value, but unlike CQL that directly estimates this quantity, conservative Q-value $Q_c$ in BRAC depends on both the estimated behavior policy $\hat{\pi}_\beta$ and the learned Q-function $Q_\theta$. Thus, we choose to apply our capacity-decreasing regularizer to both the learned Q-function $Q_\theta$ and the estimated behavior policy $\hat{\pi}_\beta$. In our experiments shown in Appendix F.2, **Figure 14 (right)**, we utilize dropout as the capacity-decreasing regularizer for BRAC and observe that it mitigates the drop in the conservative Q-value estimate and improves performance. This indicates that our workflow guideline of correcting overfitting via some form of capacity-decreasing regularization can be used to tune BRAC.

___

**Applicability to other offline RL algorithms:** We have added this discussion to **Appendix F.1** in the revised paper. While we would expect our workflow to be useful for conservative offline RL algorithms that can be characterized via Equation 2, we would not expect our workflow guidelines to be directly applicable to other algorithms that cannot be expressed with Equation 2, such as TD3 + BC, BEAR, and AWR. This is because these other algorithms utilize a "myopic" behavior constraint, i.e, while the behavior constraint in BRAC and CQL propagates through the Bellman backup, and affects Q-values, the behavior constraint in TD3+BC, BEAR, AWR only myopically constrains the learned policy to match the behavior policy at the current state, without affecting the Q-function training which resembles regular non-conservative actor-critic learning. Our workflow does not handle such algorithms, and devising a workflow for such algorithms is an interesting avenue for future work, and we now discuss this in Appendix F.1.

---

> ### Author Response · Authors · 2021-08-29
> **Answers to Common Questions of the Reviewers and the AC (Part 2/2)**
>
> ## 2. The CQL hyperparameter $\alpha$
>
> In all our experiments, we utilized a default value of $\alpha=1.0$ for the multiplier on the CQL regularizer, and this value directly follows from prior work (Singh et al. 2020). To evaluate our workflow with different $\alpha$ values, we ran additional experiments with various $\alpha=0.0, 0.01,  0.1, 2, 10, 50$ and present the results in Appendix G. We observed that our approach for detecting overfitting and policy selection is still effective with various $\alpha \in [0.1, 10)$. As shown in **Figure 15 in Appendix G**, the Q-values first increase and then decrease with more gradient steps for $\alpha=0.1, 2, 10, 50$ (in addition to $\alpha=1.0$ already in the paper). Policy selection using the peak in the average dataset Q-value returns a policy checkpoint that performs better relative to other checkpoints within the same run. Note, however, that runs with large CQL $\alpha$ values such as $\alpha=10.0, 50.0$  result in poor performance likely because a high $\alpha$ drives the learned policy to be close to the behavior policy, and our workflow fails at mitigating this. Fortunately, as we discuss next, we have now updated the paper to include a new guideline **(Guideline G.1)** that aims to detect if the value of $\alpha$ is too large, and recommends reducing it if this is the case. Our workflow is unable to improve the performance of CQL when the $\alpha$ value is very small ($\alpha \leq 0.01$). This is because CQL is unable to prevent erroneous overestimation of Q-values when $\alpha$ is small, and the policy is driven towards OOD actions as a result. To handle this, we have now updated the paper to propose a guideline to detect if $\alpha$ is too small **(Guideline G.2)** and raise it if this happens.
> ___
>
> **Detecting extremely large $\alpha$ values (Guideline G.1):** As noted above, our workflow is unable to improve the policy performance for CQL runs with extremely large $\alpha$ values that exhibit overfitting effect (per Metric 4.1). But if we can detect that a given value of $\alpha$ is quite large, we can reduce it first before applying our workflow. To this end, we now provide Guideline G.1 in Appendix G to detect situations in which the value of $\alpha$ should be decreased. This guideline proposes to compare a given run with another run with a smaller value of $\alpha$. If the run with the smaller $\alpha$ still exhibits overfitting (i.e., average dataset Q-value eventually decreases with more training), then this indicates that the original value of $\alpha$ is excessive, and we can reduce $\alpha$ to this smaller value instead, and then apply our workflow. We validate this guideline empirically in **Appendix G.1**.
>
> ___
>
> **Detecting extremely small $\alpha$ values (Guideline G.2)**: Our workflow is also unable to improve the policy performance of CQL runs with very small $\alpha$ (see Figure 17) since this leads to a catastrophic overestimation of Q-values at unseen actions. In this case, no matter how high the capacity of the model we use is, the CQL regularizer will not be minimized well and will attain large values. Building on this insight, Guideline G.2 detects if the value of $\alpha$ is small by considering the value of the CQL regularizer obtained during an attempt to correct underfitting (per Metric 4.2). If the value of $\alpha$ is large enough, and it is simply low model capacity that is preventing the regularizer from being minimized, then correcting underfitting by increasing model capacity will reduce the CQL regularizer value. However, if the value of the CQL regularizer remains unaffected when we increase model capacity, it indicates that the $\alpha$ value is too small and needs to be raised. We can then first increase $\alpha$ and then apply our workflow guidelines from the main paper. We provide a concrete example applying this procedure in Appendix G: **Guideline G.2 and Figure 18**.
>
> ___
>
> In Appendix G, we apply Guidelines G.1 and G.2 to tune $\alpha$ values for a subset of the simulated domains from Scenario #1. We will experiment with applying these guidelines to our real-world domains for the final version of the paper.

---

### Author Response · Authors · 2021-08-29
**Summary of the Main Updates to the Paper**

We thank the reviewers and the AC for their constructive and very useful feedback. We have responded to the reviewers’ and the AC’s questions, and in this global comment, we provide a brief summary of the main updates we have made to the paper to address the questions. We have added several Appendices to the paper, [that can be accessed here](https://drive.google.com/file/d/1OeHoJmRxZqMKePEOjQUSIlhlvPrQWPhZ/view?usp=sharing) or can be found in the supplementary material. The changes are indicated in $\textcolor{blue}{blue}$.


1. Added a new **Appendix F** that **(1)** discusses which offline RL algorithms our workflow is applicable to (Appendix F.1) and **(2)** empirically demonstrates an application of our workflow on to BRAC (Wu et al. 2019), a policy-constraint offline RL algorithm (Appendix F.2, results shown in Figure 14).

2. Added a new **Appendix G** to discuss the effect of the CQL hyperparameter $\alpha$ on conclusions made by our workflow. We find that our workflow is effective for a range of $\alpha$ values (Figure 15), but is unable to improve the performance of CQL when $\alpha$ is too small (Figure 17) or too large because CQL does not perform well with such values of $\alpha$. To address this, we propose two new guidelines **Guidelines G.1 and G.2** that allow us to detect and modify the $\alpha$ values if they are too large or too small, before applying our workflow for overfitting and underfitting. We empirically demonstrate the efficacy of these guidelines (Figures 17, 18).

3. Added a new **Appendix I** that evaluates various capacity-decreasing regularizers ($l_1$, $l_2$ regularization and dropout) for overfitting correction (results shown in Figure 20)

4. Added a new **Appendix H** that evaluates the proposed underfitting correction for 10 and 20 objects in Scenario #2 (results in Figure 19)

5. Added a new **Appendix J** to compare the evolution of the overfitting metric suggested by Reviewer D6Ls and Metric 4.1. We find that both metrics are expected to return identical policy checkpoints and will thus perform similarly for policy selection (results in Figure 21).

In addition, we have updated the main paper to address other clarity questions and typos, and have added pointers to the respective appendices to clearly point the reader to the relevant experimental study in the Appendix.

---

### Meta-Review · Area_Chair_hnKZ · 2021-08-06

**Recommendation:** Accept (Oral)
**Confidence:** 4

**Metareview:**

The paper studies an important topic of accessing overfitting and underfitting in CQL.

All reviewers see merits in this work but also point out weaknesses and have concerns that I share.
I agree with the evaluation of the reviewers and invite the authors to answer the raised questions and indicate how the paper will be improved for the camera-ready version.
In particular: (however, this does not mean the other points should not be addressed)
 - Scenario #2 regarding the object 35 task and alternative tasks
 - How much does the CQL hyperparameter alpha affect the conclusions? For the overfitting case, would alpha affect the drop of the Q-value?
 - The relation and applicability to other offline RL algorithms

The authors have addressed most of the concerns and have added several additional experiments.
I recommend acceptance of the paper.

---

> ### Comment · Area_Chair_hnKZ · 2021-08-27
> **Waiting for Author response**
>
> Dear Authors,
>
> please respond to the reviewer's questions by the 30th of August.
>
> Best regards,
>
>  your AC

---

> > ### Author Response · Authors · 2021-08-29
> > **Apologies for the delay, we have responded to the questions**
> >
> > Dear AC,
> >
> > We sincerely apologize for the delay in responding to reviewers' questions. We have responded to these questions now by updating the paper with new **Appendices F-J** that present various experiments for answering the questions. For the common concerns of reviewers, we have provided a detailed answer in the form of a global comment and a discussion in the Appendices.
> >
> > We have also listed down a summary of updates made to the paper in the form of a global comment.
> >
> > Thank you very much!
> >
> > Best,
> > Authors

---

> ### Author Response · Authors · 2021-08-29
> **Author Response**
>
> We thank the area chair for their constructive feedback. We have responded individually to the reviewers in our responses below and answered their questions by updating the paper with experiments in the Appendices. Here, we briefly summarize how we have addressed the questions marked in bulleted points in the meta-review:
>
> **Relation and applicability to other offline RL algorithms:** We provide a detailed response to this question in our [global comment to all the reviewers and the AC](https://openreview.net/forum?id=fy4ZBWxYbIo&noteId=7CIFzgxtrC) since this was a common concern. We leave the details to the global comment and the section in **Appendices F.1 and F.2** and briefly, summarize our findings here. Our workflow is applicable to offline RL algorithms that can be expressed as optimizing Equation 2, which we refer to as ``conservative offline RL’’ algorithms. Policy constraint algorithms such as BRAC (Wu et al. 2019), also admit a similar optimization objective, and thus, **we applied our workflow to BRAC (Wu et al. 2019) in Appendix F.2.** We find that in experiments on the simulated drawer task with 100 and 200 trajectories from Scenario #1 in the paper, conservative Q-value estimates from BRAC can be used to detect overfitting and to perform policy selection. We can then alleviate this overfitting (i.e., drop in average dataset conservative Q-value) by adding dropout, a capacity-decreasing regularizer and this leads to improved policy performance as shown in Figure 14 (right) in Appendix F.2.
>
> We have added updated **Appendix F.1** to discuss that our workflow is not applicable for other offline RL algorithms such as TD3+BC, BEAR, AWR, that do not fit the form of the conservative objective in Equation 2, which is required for our approach. Devising a workflow for these algorithms is an interesting avenue for future research.
>
> ___
>
> **Scenario #2 regarding object 35 task, and alternative tasks:** We have now updated the paper to include experiments with 10 and 20 objects for the underfitting Scenario #2 in **Appendix H, Figure 19.** We observe that in cases where the training TD error is high (i.e., in the case of 10 and 20 objects), utilizing a ResNet policy + DR3 capacity-increasing regularizer for alleviating underfitting leads to improved performance over naive CQL as shown in Figure 19. Additionally, we also observe that the capacity-increasing regularization that we apply reduces the value of training TD error for both 10 and 20 objects (Figure 19). This validates our underfitting hypothesis.
>
> ___
>
> **Effect of CQL hyperparameter $\alpha$ on conclusions:** In all our experiments, we utilized a single default value of $\alpha=1.0$ for the multiplier on the CQL regularizer drawn from prior work (Singh et al. 2020). Since this was a common question, we provide a detailed response in our [global comment to the Reviewers and the AC](https://openreview.net/forum?id=fy4ZBWxYbIo&noteId=dU6k2lKPEIF). To briefly summarize our findings, we found that our workflow is robust across several $\alpha$ values, but is unable to improve CQL when $\alpha$ is too large or too small. Note that CQL itself performs badly for these $\alpha$ values as an extremely small $\alpha$ fails to prevent catastrophic overestimation in Q-values and an extremely large $\alpha$ heavily regularizes the learned policy towards the behavior policy.
>
> However, we have now updated the paper with two new guidelines **(Guidelines G.1 and G.2)** in **Appendix G** to handle these scenarios by first detecting if $\alpha$ is too small or too large and then modifying the value of $\alpha$ accordingly. After the value of $\alpha$ is modified, we are able to apply the workflow presented in the main paper to tune and improve the performance of CQL. Details including an empirical validation are provided in Appendix G, and our global comment is linked [here](https://openreview.net/forum?id=fy4ZBWxYbIo&noteId=dU6k2lKPEIF).

---

### Decision · Program_Chairs · 2021-09-13

**Decision:**

Accept (Oral)

**Comment:**

The paper studies an important topic of accessing overfitting and underfitting in CQL.

All reviewers see merits in this work but also point out weaknesses and have concerns that I share.
I agree with the evaluation of the reviewers and invite the authors to answer the raised questions and indicate how the paper will be improved for the camera-ready version.
In particular: (however, this does not mean the other points should not be addressed)
 - Scenario #2 regarding the object 35 task and alternative tasks
 - How much does the CQL hyperparameter alpha affect the conclusions? For the overfitting case, would alpha affect the drop of the Q-value?
 - The relation and applicability to other offline RL algorithms

The authors have addressed most of the concerns and have added several additional experiments.
I recommend acceptance of the paper.